Enhancing market trend prediction using convolutional neural networks on Japanese candlestick patterns

Mersal Edrees Ramadan 1
Karaoğlan Kürşat Mustafa 1
Kutucu Hakan 2 hakankutucu@karabuk.edu.tr
1 Department of Computer Engineering, Karabuk University , Karabuk , Turkey
2 Department of Software Engineering, Karabuk University , Karabuk , Turkey
Cunkas Mehmet
Electronic publication date: 2025 Feb 27
Publication date: 2025
Volume: 11
Electronic Location ID: e2719
Received 2024 Aug 20; Accepted 2025 Jan 28
Copyright: © 2025 Mersal et al.
Copyright year: 2025
Copyright holder: Mersal et al.
License: This is an open access article distributed under the terms of the Creative Commons Attribution License, which permits unrestricted use, distribution, reproduction and adaptation in any medium and for any purpose provided that it is properly attributed. For attribution, the original author(s), title, publication source (PeerJ Computer Science) and either DOI or URL of the article must be cited.
License URL: https://creativecommons.org/licenses/by/4.0/

Keywords: Stock market, Buy-sell strategy, CNN, Cross-validation, Japanese candlestick

Funding: The authors received no funding for this work.

==============================
This study discusses using Japanese candlestick (JC) patterns to predict future price movements in financial markets. The history of candlestick trading dates back to the 17th century and involves the analysis of patterns formed during JC trading. Candlestick patterns are practical tools for the technical analysis of traders in financial markets. They may serve as indicators of traders’ documents of a potential change in market sentiment and trend direction. This study aimed to predict the following candle-trend-based JC charts using convolutional neural networks (CNNs). In order to enhance the accuracy of predicting the directional movement of subsequent financial candlesticks, a rich dataset has been constructed by following a structured three-step process, and a CNN model has been trained. Initially, the dataset was analyzed, and sub-charts were generated using a sliding window technique. Subsequently, the Ta-lib library was used to identify whether predefined patterns were present within the windows. The third phase involved the classification of each window’s directional tendency, which was substantiated by employing various technical indicators to validate the direction of the trend. Following the data preparation and analysis phases, a CNN model was developed to extract features from sub-charts and facilitate precise predictions effectively. The experimental results of this approach demonstrated a remarkable predictive accuracy of up to 99.3%. Implementing cross-validation techniques is essential to verify the reliability and overall performance of the model. To achieve this goal, the dataset was divided into several small subsets. Subsequently, the model was trained and evaluated multiple times using different combinations of these subsets. This method allows for a more accurate assessment of the model’s predictive capabilities by examining its performance on unseen data.

Introduction

Forecasting anticipates events that occur in the near or distant future by applying the concept of forecasting to financial markets, which means determining the direction of market movement. Many investors, traders, and companies strive to achieve high forecasting accuracy to make decisions that increase revenues or avoid losses. Despite the use of numerous tools to predict the direction of the next candle, this issue still needs to be solved. This is due to the dynamic nature of financial markets, which constantly fluctuate between rising and falling prices (Dakalbab et al., 2024). Because political and economic factors differ from country to country. Information obtained or disseminated among traders about corporate assets, the psychological state of traders, and other factors can influence traders’ feelings and fears, changing the overall market trend and causing a downtrend or uptrend.

Currently, the development of Artificial Intelligence (AI) systems has significantly increased in all fields, leading to the desire to exploit this potential to predict financial market movements. An abundance of historical data has reinforced this desire. Therefore, researchers and scholars in this field constantly seek to study historical data on financial markets using technical analysis to predict upcoming market fluctuations and enhance the accuracy of their forecasts by developing new tools, techniques, and methodologies. This strategy relies on historical data, specifically time-series data that include the opening and closing prices of securities, the highest and lowest prices reached, and trading volume. Furthermore, it involves the use of technical indicators, which graphically represent price patterns over a specific time frame, and Japanese candlesticks (JC), which visually depict price movements. An essential aspect of predicting future price movements is the analysis of the patterns formed during JCs trading. These patterns are consistently repeated across historical data, regardless of price variations (Mersal & Kutucu, 2024). The history of JCs dates back to the 17th century when rice trader Munehisa Homma traded in the futures market and discovered that traders’ emotions, along with demand and supply, could significantly influence rice prices (Chen & Tsai, 2022; Dakalbab et al., 2024). This led him to develop a method to represent market price movement. The candlestick shown in Fig. 1 contains four main information components. These components are the opening and closing prices and the high and low prices during specific trading periods. The center part of the candle, known as the body, is formed between the opening and closing prices. The shadow or wick above or below the body represents the high and low prices, respectively (Hung et al., 2020). Initially, there was no color coding of candlesticks, but color coding has been adopted in the modern era. Red and black represent bearish candles, while green and white represent bullish candles. In the 1980s, Steve Nison introduced the candlestick chart to the Western world (Chen & Tsai, 2020).

Figure 1 The components for bullish and bearish candles.

Although candlestick formations can be analyzed individually, they are primarily scrutinized when related as part of a group. These groupings include patterns in which numerous candlesticks come together to describe a shape or configuration, which helps traders interpret them more concisely (Hung et al., 2020). Many of the patterns look like geometric patterns or objects that can be familiar to the eye. For example, a candlestick whose formation looks like a hammer can be called a hammer, or one with a relatively equal opening or closing price can be called a Doji. These candlestick formations may serve as indicators showing traders documents of a potential change in market sentiment and trend direction, paving the way for informed decisions on asset purchases or sales. Doji pattern formation is an excellent example of when buyers’ and sellers’ sentiments are balanced, which can lead to a potential shift in a given trend. Thus, candlestick patterns are an effective tool for the technical analysis of traders in financial markets. Table 1 lists a set of JCs patterns indicating a bullish market. Table 2 contains a JCs pattern collection that acknowledges a bearish market (ChartSchool, 2024). Patterns give traders ideas about pricing or have implications for market sentiment, price moves, or the behavior of traders. Knowing what this implies helps decide when a reversal or trend continuation will occur. The main aim of this study is to predict the next candle trend using JCs charts based on deep learning classification and convolutional neural networks (CNN).

Table 1 Bullish candlestick patterns: structural examples and components of candle types.

Abandoned baby	Belt-hold	Breakaway	Closing marubozu	
				
Concealing baby swallow	Counterattack	Doji Star	Dragonfly Doji	
				
Engulfing	Hammer	Harami	Harami Cross	
				
Homing Pigeon	Inverted Hammer	Kicking	Ladder Bottom	
				
Long Line Candle	Marubozu	Mat Hold	Matching Low	
				
Morning Doji Star	Morning Star	Piercing Line	Rising Three Metdods	
				
Separating Lines	Side by Side White Lines	Stick Sandwich	Takuri	
				
Tasuki Gap	Three Inside Up	Three Line Strike	Three Outside Up	
				
Three Stars in The Soutd	Three White Soldiers	Tri-Star	Unique Three-River	
				
Upside Gap Three Metdods	
	

Table 2 Bearish candlestick patterns: structural examples and components of candle types.

Abandoned baby	Advance block	Belt-hold	Breakaway	
				
Closing Marubozu	Counterattack	Dark Cloud Cover	Deliberation	
				
Doji Star	Downside Gap Three Methods	Engulfing	Evening Doji Star	
				
Evening Star	Falling Three Methods	Gravestone Doji	Hanging Man	
				
Harami	Harami Cross	In Neck	Kicking	
				
Long Line Candle	Marubozu	On Neck	Separating Lines	
				
Shooting Star	Side by Side White Lines	Tasuki Gap	Three Black Crows	
				
Three Inside Down	Three Line Strike	Three Outside Down	Thrusting	
				
Tri-Star	Upside Gap Two Crows	
		

Many factors, such as general economic conditions and political or financial events, directly or indirectly affect the development of financial time series. However, predicting the stock market using candlestick charts is costly and laborious. This article proposes an approach to analyze patterns using a candlestick chart with technical indicators to predict financial trends. The Ta-lib library generates candlestick charts to capture 61 patterns in financial time series. The resulting patterned charts are classified using technical indicators for bullish and bearish trend predictions. The analysis methodology leverages both the high-order properties of candlestick patterns and the recurring formations observed in financial time series. A three-step method is presented, such as detecting patterns in historical data for the Forex 15-M stock market and creating windows with a sliding window from these charts, classifying all windows as bullish and bearish based on the simple moving average (SMA), and inputting the candlestick chart data into the CNN to predict the trend.

This research presents a methodological advancement in financial forecasting by proposing an integrated deep learning framework for candlestick pattern analysis. The methodology addresses current limitations in market prediction systems through three key innovations: (1) the comprehensive implementation of 61 distinct candlestick patterns within a CNN architecture, significantly expanding upon previous studies that typically utilize fewer patterns, (2) the development of an optimized window-shift mechanism for high-frequency Forex data analysis, and (3) the integration of SMAs for enhanced pattern labeling accuracy. The empirical results, demonstrating 99.3% accuracy in trend prediction, suggest substantial improvements over existing methodologies documented in the literature, which typically achieve accuracy rates between 56% and 91.51%. This quantitative advancement is particularly significant for high-frequency trading environments, where traditional pattern recognition methods often face limitations due to market volatility and noise. Furthermore, the study’s systematic evaluation of various temporal resolutions (window sizes: 5–30, with corresponding shifts) provides a robust framework for analyzing the relationship between pattern recognition accuracy and temporal scale in financial markets. These methodological contributions enhance both the theoretical understanding of market pattern dynamics and the practical implementation of automated trading systems.

This research presents a novel approach for predicting directional movements in financial markets using candlestick patterns. The key contributions are as follows: Implementation of a comprehensive pattern recognition system incorporating 61 distinct candlestick patterns within a CNN architecture. The integration of the Ta-lib library enables automated pattern detection and classification, significantly expanding upon existing approaches that typically analyze 10–15 patterns. This extensive pattern coverage enhances the model’s capability to identify complex market formations.

Development of an adaptive window-shift mechanism optimized for high-frequency Forex data analysis. The methodology systematically evaluates six temporal configurations (window sizes: 5, 10, 15, 20, 25, 30) with corresponding shift parameters of 50%, establishing a framework for multi-scale pattern recognition while preserving temporal relationships in the data.

Design of a quantitative trend identification framework that integrates SMAs with candlestick patterns. This methodology establishes objective criteria for pattern labeling and trend classification. Empirical evaluation demonstrates classification accuracy of 99.3% in binary trend prediction, compared to existing methodologies which achieve accuracy rates between 56% and 91.51%.

Validation of model robustness through stratified k-fold cross-validation (k = 5) on EUR/USD forex pair data sampled at 15-min intervals. The evaluation framework incorporates multiple market conditions and temporal scales, demonstrating the model’s generalization capabilities and practical applicability in real-time trading environments.

In summary, this research advances the field through four key methodological contributions: (1) the integration of SMAs with candlestick patterns for quantitative trend labeling, (2) the comprehensive implementation of 61 distinct candlestick patterns for enhanced pattern recognition, (3) systematic evaluation of multiple temporal resolutions through varied window configurations, and (4) empirical validation achieving 99.3% accuracy in binary trend classification. These contributions establish a robust framework for automated pattern recognition in high-frequency forex trading environments.

Following this introduction, we present a review of related work in candlestick pattern recognition and deep learning-based market prediction approaches. We then detail our materials and methodology, encompassing dataset preprocessing, candlestick chart creation, pattern identification, dataset labeling, model architecture, and evaluation framework. Subsequently, we present our experimental results and statistical analysis, followed by a discussion of comparative analysis, model complexity, and real-time market performance. The article concludes with research limitations, validity considerations, and data availability.

Related work

This section discusses the primary research studies available in the literature on the topic of our study and advances in this field. These studies provide a baseline of existing knowledge and research gaps.

Nguyen et al. (2023) investigated how well JCs patterns could predict stock trends. They used an object detection technique to examine chart patterns, identify them, and test the accuracy of the price-action candles. The image data included a variety of patterns, such as candlesticks, head and shoulders, reverse head and shoulders, and double tops and bottoms. This essay discusses a novel combination of contemporary object detection methods with GAF time series encoding to analyze candlestick patterns (Nguyen et al., 2023).

Chen & Tsai (2022) made significant changes to the original YOLO version 1 model using a time eries encoding technique by leveraging the characteristics of this data type. The proposed model, utilizing deep neural networks and a unique architectural design, demonstrated its abilities for candlestick categorization and location recognition (Chen & Tsai, 2022).

Santur (2022) developed a framework using candlestick charts to predict trend direction. The study consisted of four stages. Initially, a candle-pattern recognition system was developed. In the second phase, the performance of the model was evaluated by conducting training and testing procedures using datasets that included labeled candlestick chart types and trend directions. During the machine learning phase, community approaches, such as XGBoost, were employed. During the last phase of the study, it was observed that employing a strategy that solely relies on identifying candlestick patterns and taking positions in line with the trend direction of the proposed approach led to greater profits in 11 global indices compared with the buy-and-hold strategy (Santur, 2022).

Heinz et al. (2021) examined the effectiveness of bullish engulfing and bearish engulfing patterns. He emphasized the possibility of predicting, in the short term, a bullish engulfing pattern with low criteria and no closing price criterion. Similarly, predict the bearish engulfing pattern in the short term based on the open and high criteria but not on the close price criteria (Heinz et al., 2021). According to Cohen (2020), the pattern “stairs” the developed can generate profits in all 20 stocks tested, outperforming the buy-and-hold strategy, which was only profitable in 16 of the 20 stocks. The results obtained in testing the “Engulfing” pattern did not achieve positive gains, and the “Harami” pattern barely achieved success. The complex “Kicker” pattern outperformed the buy-and-hold strategy with an average positive profit (Cohen, 2020).

Ramadhan, Palupi & Wahyudi (2022) proposed the use of a CNN-LSTM model to assess the financial performance of trading positions in the stock market based on candlestick patterns. As part of their study, they showed the candlestick patterns as pictures and used CNN and the Angular Field (AF) and Gramian Angular Field (GAF) techniques to find certain candlestick configurations accurately. They also used the long short-term memory (LSTM) model to predict future stock prices. The test results show that the CNN-LSTM model performs better than the CNN model in predicting profitable trading positions. It even gets it right 82.7% of the time when it comes to figuring out how long to hold profitable candlestick patterns for 3 h (Ramadhan, Palupi & Wahyudi, 2022).

Chen & Tsai (2020) presented a method to classify candlestick patterns in financial markets By using the GAF to encode time series data into images, a model was designed using CNN to recognize important candlestick patterns with high accuracy. The model performs well in recognizing patterns, which is better than that of any traditional LSTM model. This demonstrates the usefulness of picture-based financial time series data representation (Chen & Tsai, 2020).

However, Xu (2021) provided evidence that the use of deep learning models is not a better option for recognizing candlestick features based on GAF images. This study demonstrates this by comparing the performances of the MLP, CNN, AdaBoost, RF, and XGBoot models with GAF images. The results showed that MLP and CNN were better than RF and AdaBoost but not better than XGBoost (Xu, 2021).

According to Cagliero, Fior & Garza (2023) trading systems based on machine learning tend to generate many false signals and do not adequately consider the information provided by candlestick patterns to overcome these effects. Therefore, they proposed separating the machine learning and pattern recognition steps so that the trading system could generate fewer double-check recommendations. To achieve this goal, researchers have explored methods that combine machine learning and pattern recognition strategies, including supervised deep learning and autoregressive techniques (Cagliero, Fior & Garza, 2023).

Karmelia, Widjaja & Hansun (2022) used the feed-forward neural network to classify candlestick patterns found in historical data. A challenge was the problem of data imbalance, which had an impact on the classification accuracy, as they used techniques such as random undersampling and SMOTE. The model showed high classification accuracy, but the F1 score was below a low-level (Karmelia, Widjaja & Hansun, 2022).

Orquín-Serrano (2020) developed a novel classification method to effectively evaluate the ability of JCs patterns to predict forex market movements for the EUR–USD pair. This method categorizes individual candlestick patterns based on adaptive criteria. assessed this using a statistical inference method that considered a component of the average rate of strategic returns in historical data that was not included in the statistical sample. However, even after considering transaction costs, no net positive returns were observed using this technique (Orquín-Serrano, 2020).

Hung & Chen (2021) proposed a deep price–movement prediction model that uses candlestick charts in historical data. The model was built in three stages. The first stage involved preparing data by analyzing the JCs chart into a group of subcharts. The second stage involved selecting the best representation of the sub-graphs using a CNN autoencoder. Finally, an RNN was applied to predict the price movements (Hung & Chen, 2021).

Birogul, Temur & Kose (2020) proposed a real-time object recognition system using the You Only Look Once (YOLO) algorithm to identify “buy-sell” objects in 2D candlestick charts. They proposed that traders use these buying and selling components to determine the optimal timing for purchasing or selling an investment instrument. To make accurate predictions, this study also emphasizes the importance of adopting a distinct perspective while analyzing charts in real-time, considering the visual patterns observed in the historical trends of the investment tool (Birogul, Temur & Kose, 2020).

The study by Lin et al. (2021) introduced a pattern recognition model (PRML) to enhance financial decisions. The results show that 2-day candlestick patterns are more effective for forecasting than other machine learning models. The PRML provides a profitable investment strategy. This study applied PRML to five strategy pools, revealing that 2-day candlestick patterns have the most predictive power (Lin et al., 2021).

Brim & Flann (2022) aimed to surpass S&P 500 index returns by integrating a CNN with a double deep Q-network (DDQN). Trained a CNN to interpret candlestick charts and generate feature map representations to enhance our understanding of market trends. To maximize returns, the DDQN component optimizes trading strategies and executes trades. This demonstrates that combining deep- and reinforcement-learning techniques can create more effective trading systems. CNN’s ability of CNNs to focus on recent candles demonstrates advanced analytical precision (Brim & Flann, 2022).

Hung et al. (2020) employed candlestick charts as data sources, the Deep Candlestick Predictor (DCP) is an innovative framework that employs the CNN methodology to forecast stock prices. The DCP divides a chart into subcharts using a chart decomposer, which then extracts local patterns and encodes them using a CNN autoencoder. To improve the prediction accuracy later implemented the 1D-CNN in price movement forecasting. The test results for the Taiwan Exchange Capitalization Weighted Stock Index showed better accuracy than traditional methods. This shows that using visual data from candlestick charts can improve predictions of financial markets (Hung et al., 2020).

Chen & Tsai (2022) exposed an Adversarial Generator to enhance and explain the model, slightly altering the input values to create a model of candlestick patterns that closely resemble traders’ expectations. Candles were originally a visual tool for traders to help them understand price movements. A CNN can identify and define patterns in candles. However, there is still a mystery regarding how they arrived at their conclusion. Therefore, this study focused on creating a model that explains the decision-making process more clearly while identifying specific candlestick patterns (Chen & Tsai, 2022).

Liang et al. (2022) used sequence similarity and JCs patterns to develop a novel method for predicting stock market trends. In light of the intricacy and challenges with interpretation associated with multimodal data in financial forecasting, this research suggests a simpler approach based on JCs data. This process consisted of two steps. Initially, sequential pattern mining was used to extract candlestick patterns from multidimensional data. introduced a new sequence similarity measure for aligning candle sequences with known patterns (Liang et al., 2022).

Madbouly et al. (2020) introduced a model combining the cloud model, fuzzy time series, and Heikin-Ashi candlesticks to accurately predict stock market trends. This innovative method addresses stock market challenges, such as uncertainty, nonlinearity, and noise, by merging Heikin-Ashi candlesticks, which clarify trends by averaging past and present prices. The cloud model navigates qualitative and quantitative data gaps by crafting membership functions to manage the ambiguity of historical data. The model forecasts future stock prices and trends in dynamic weighted fuzzy logical relationships and has a high forecasting accuracy in empirical evaluations (Madbouly et al., 2020).

Chen, Hu & Xue (2024) introduced a stock price predictor system called the Sparrow Search Algorithm Candlestick patterns bidirectional gated recurrent unit (SSA-CPbiGRU). This system combines candlestick patterns with a Sparrow Search Algorithm (SSA). Using candlestick patterns, the input data are provided with structural characteristics and time-series relationships. In addition, the hyperparameters of the CPBiGRU model were optimized using SSA. These fine-tuned hyperparameters are then utilized in the network model to generate precise forecasts. They performed experiments on six stocks chosen from six distinct sectors of the Chinese stock market. The experimental results demonstrated a substantial enhancement in prediction accuracy. On the median, the proposed approach achieved 31.13% less test loss in terms of mean absolute percentage error (MAPE), 24.92% less test loss in terms of root mean square error (RMSE), and 30.42% less test loss in the context of MAE. Furthermore, the suggested model showed an average enhancement of 2.05% in R2 (Chen, Hu & Xue, 2024).

Lin et al. (2021) developed a machine-learning ensemble prediction model that could automatically select the best prediction methods concerning daily k-line patterns. This model made use of AI techniques and traditional candlestick charts, The study was based on Taoist cosmology and the eight-trigram classification and incorporated insights on 2-day trading, comprised of high and low prices. This research suggested a methodology to find the most efficient machine prediction method among many feature modes to amalgamate numerous prediction methods. The ensemble model balanced the parameter optimization to uplift the top six predictive model efficiency, specifically random forest (RF), gradient boosting decision tree (GBDT), logistic regression (LR), and k-nearest neighbors (KNN), and support vector machine, and long short-term memory (LSTM). The experimental results revealed that RF and GBDT shorted on both KNN and SVM had a much better performance in the short term. The investment strategy based on the performance could beat buy and hold investment in the investment stock, including reducing the maximum drawdown and Sortino index, and improving the Sharpe index (Lin et al., 2021).

To forecast prices of crypto assets traded on futures markets, Orte et al. (2023) introduced a model that is implemented with a random forest. They used input variables to have three different alternatives: technical indicators, candlestick patterns, or a mix of both simultaneously. Later, they also dealt within the model parameters, optimal time intervals, and investment horizons. Finally, during their presentation, they showed the model’s output and conducted a 1-year out-of-sample prediction, noting that they have chosen the whole 2020 as the observation period due to three potential scenarios seen in the stock market in this year. There was an uptrending market, and bearish due to the global pandemic and a sideways market. To make it look closer to reality, the researchers have retraced the model on each collection dataset to ensure it always has the most current information. in summary, candlestick patterns instead of technical indicators have proven to be a key to significantly improving the model results (Orte et al., 2023). Barra et al. (2020) applied an ensemble of CNNs trained over GAF images which were generated over time series concerning the Standard & Poor’s 500 index future. The goal is to determine the future trend of the U.S. market. To obtain input for CNN, multi-resolution imaging was which is used to analyze different time intervals. In the experiment, the proposed model consistently outperforms the B&H strategy. Both quantitative and qualitative results are provided (Barra et al., 2020).

Material and methodology

The primary goal of the current study is to increase the precision of future trend prediction in the financial market by combining the traditional methodology of candlestick pattern charting with the contemporary capabilities of deep learning classifications, such as CNNs. More specifically, the highly volatile Forex financial market was used in the study; a dataset was carefully selected from the continuous over-covered period from February 21, 2020, to February 23, 2024, with a time frame of 15 m. However, as far as financial market data are concerned, it is clear that it is not random and is both sequential and rich with different patterns. Figure 2 shows a flowchart of the proposed system. Time-series data were transformed into candlestick pattern charts.

Figure 2 Flowchart of whole proposed system.

This study uses a more advanced method called sliding window (Guo & Wen-jing Li, 2023), as shown in Fig. 3 flowchart, to create a candlestick chart and classify it as an uptrend or a downtrend. The process segments the entire dataset systematically and forms a series of different, picture-like ‘sub-images’; a fixed number of candlesticks is presented in each ‘sub-image.’ The formation of the ‘sub-image’ starts and ends are also defined very carefully to ensure similar data arrangement in all of the ‘sub-images’. This step is used not only to extract the pattern of the candlesticks in a more accurate way, but also to present the data in a way that suits CNN’s ability to versatilely recognize and analyze visual data recognize and analyze visual data versatilely.

Figure 3 Flowchart for labeling and creating candlestick charting.

Dataset

The data for this study is taken from Forex Historical Data provided by the Forex Strategy Builder website (https://forexsb.com/historical-forex-data). This repository offers a comprehensive dataset that includes various time frames for currency pairs, focusing specifically on Euro and dollar (EUR/USD) transactions. The dataset was carefully selected from the continuous over-covered period from February 21, 2020, to February 23, 2024, with a time frame of 15 m. The dataset encompasses a rich collection of time series data, which contain opening and closing prices along with the highest and lowest prices for each trading session. In addition, it provides information on the trading volumes. The data preprocessing methodology addressed three critical aspects. First, missing values in price and volume data were handled through forward filling for isolated instances and record removal for consecutive gaps. Second, we implemented filtering criteria to ensure the completeness of Open, High, Low, Close (OHLC) values for accurate candlestick representation. Third, outlier detection was performed using z-score analysis, with extreme values validated against documented market events to maintain data integrity while preserving legitimate price movements.

Creation of candlestick charts

The candlestick charts were created using a sliding window; each time, a set of time-series data was selected and converted into a candlestick chart. The sliding window technique divides the dataset into subsets or “windows,” with a set length and slides them systematically at a step across the dataset. Every window contains a sequence of points, allowing for short-term analysis and pattern identification. Key parameters include the following: Window size determines the number of data points included; the larger the number of points, the longer the trend time considered, and the smaller size. The scroll size defines the number of points the window moves through in each iteration; when the step size is smaller than window size, the overlap between windows occurs, making analysis more continuous.

Figure 4 shows the process of selecting data using a sliding window and creating candlestick charts with patterns. In this study, 5, 10, 15, 20, 25, and 30 window sizes were tested to find the most appropriate set of candles containing the patterns for each sub-graph. Determining the shift size is extremely important for the model’s training. For a single scroll size, a significant part of the exact window overlaps, and the difference in each sub-graph is one unit candle; if half the window size is chosen, the difference is halved, or if the window size is chosen precisely, the sub-graph will be completely different.

Figure 4 The illustration of the sliding window.

Identification of candlestick pattern

After selecting the data using a sliding window, we determined whether they contained a pattern. Candlestick patterns are of significant importance in technical analysis, particularly because they mark the patterns of market conditions and potential price developments. Specifically, there are slightly over three allocated candlesticks; however, our research scrutinizes strategic patterns for 61 candlesticks. This framework is explained by the functionality of the Technical Analysis Library, which is viewed as one of the most widely used instruments for comprehensive support in analyzing financial markets. The respective candlesticks include: “Two Crows”, “Three Black Crows”, “Three Inside Up/Down”, “Three-Line Strike”, “Three Outside Up/Down”, “Three Stars In The South”, “Three Advancing White Soldiers”, “Abandoned Baby”, “Advance Block”, “Belt-hold”, “Breakaway”, “Closing Marubozu”, “Concealing Baby Swallow”, “Counterattack”, “Dark Cloud Cover”, “Doji”, “Doji Star”, “Dragonfly Doji”, “Engulfing Pattern”, “Evening Doji Star”, “Evening Star”, “Up/Down-gap side-by-side white lines”, “Gravestone Doji”, “Hammer”, “Hanging Man”, “Harami Pattern”, “Harami Cross Pattern”, “High-Wave Candle”, “Hikkake Pattern”, “Modified Hikkake Pattern”, “Homing Pigeon”, “Identical Three Crows”, “In-Neck Pattern”, “Inverted Hammer”, “Kicking”, “Kicking-bull/bear determined by the longer marubozu”, “Ladder Bottom”, “Long Legged Doji”, “Long Line Candle”, “Marubozu”, “Matching Low”, “Mat Hold”, “Morning Doji Star”, “Morning Star”, “On-Neck Pattern”, “Piercing Pattern”, “Rickshaw Man”, “Rising/Falling Three Methods”, “Separating Lines”, “Shooting Star”, “Short Line Candle”, “Spinning Top”, “Stalled Pattern”, “Stick Sandwich”, “Takuri (Dragonfly Doji with very long lower shadow)”, “Tasuki Gap”, “Thrusting Pattern”, “Tristar Pattern”, “Unique 3 River”, “Upside Gap Two Crows”, “Upside/Downside Gap Three Methods”.

Dataset labeling methods

The labeling of candlestick charts can significantly impact the interpretation and resultant strategies derived from financial market analysis.

The first method for labeling candlestick charts is comparing market prices in a sequence. The idea is to compare the closing price of one candlestick directly with the closing price of another. this method with a simple strategy that identifies trends on a limited scale and pays close attention to the amplitude and direction of prices, which way up and which way down. The second method for labeling is measuring the percent change, since strategy two is helpful in volatile market periods as it tracks the percent change between candlesticks. The third method is to label by developing strategies to identify candlesticks. It extends beyond just comparing the prices on the market. With this strategy, can implement statistical and machine learning techniques to cluster the candlesticks based on past data and predict possible future trends.

The labeling process occurs immediately after pattern recognition and involves classifying each identified pattern present in the historical dataset as a signal of either an uptrend or downtrend. The classification process is based on the pattern’s performance in similar market conditions, with more weight placed on additional technical indicators, such as moving averages and RSI levels for context. Indeed, technical indicators serve as confirmatory signals because a pattern can suggest a trend change but is completely wrong. In contrast, a pattern can be entirely correct but insufficient for validating a reversal or continuation. This process ensures that we have the critical framework necessary for predicting future trends and the corresponding confirmation required to train the models. This methodology ensures that the models generate highly reliable predictions with real-world applications for traders and analysts seeking to make highly accurate predictions of market movement. In this study, we utilized one technical indicators: the SMAs and the which is used to compare momentum changes and given its formula in Eq. (1).

SMA: This is the average of the closing prices for a given period.

(1) SMA(M,n)=1n∑k=i+ni+n−1M(k).

Prediction model

We developed a CNN model using TensorFlow and Keras to effectively classify the images into two binary classes. The dataset was divided into training, validation, and test sets in proportions of 70%, 10% and 20%, respectively. First, the images were preprocessed and augmented. In our case, the ImageDataGenerator class proved to be a powerful tool in Keras, created to assist with image augmentation during real-time processing. The class allowed us to apply rescaling and other techniques, such as shear transformation and zooming, to images. Moreover, augmentations have several benefits: they artificially expand the dataset, introduce various conditions that our images can face in real life, and prevent our model from learning noise and specific details by providing diversity to input images. The class also rescales our images, which means that it normalizes pixel values to the range of [0,1], making the pixel distributions across all input features uniform in magnitude and reach, facilitating the model to converge while training.

The CNN architecture we used was specifically designed to make it possible to process images through successive layers, and each layer is especially good at distilling separate features and decreasing the dimensionality of data while maintaining the features necessary for classification. The first layer of the multilayer network is the input layer. The input layer is designed to work with images with a size of 150 × 150 pixels, and each pixel is three-dimensional, auxiliary depth to the RGB color system. Figure 5 illustrates the structured model and the sequence of the layers.

Figure 5 The CNN architecture representing the sequential processing of an input image.

After the input layer, the first convolutional layer conv2d used 32 filters of size 3 × 3 to perform the convolution operation of the input layer. Each filter was slid over the image, and the dot product was calculated. This operation produces a feature map that enhances certain features of the images, such as sharp edges of objects and textures, while filtering out other features. The result is that the raw image data are transformed into abstracted feature maps that are more meaningful for analysis than the raw pixel values. Following the conv2d layer, a max_pooling2d layer is used for spatial pooling. The dimensionality of each feature map is reduced in the pooling process, while critical information is retained to simplify the image of each window. A 2 × 2 pooling was used; therefore, so this did the effect of summarizing the feature map into a pool of 74 × 74. The goal of pooling is to reduce the computational complexity of a network. conv2d_1 and conv2d_2 are then used to further refine the extracted features by performing deeper convolutions on the feature maps and with more filters than the previous ones. At this time, 64 and 128 filters were used. Two more max-pooling layers follow each of the identified layers, which decreases the feature map dimensions from 72 × 72 to 36 × 36 to 34 × 34 to 17 × 17. These layers look deeper into the feature maps learned from the previously used layers and capture complex patterns, such as parts of objects within the image. The pooling layers also reduce overfitting and provide an abstracted form of the features.

The flattened layer follows the last pooling layer and maps the 3D feature maps into a 1D feature vector. This pre-classification is essential to ensure that highly intricate feature representations are better processed during classification through the fully connected layers. The first dense layer is dense, contains 512 neurons, and is equipped with a rectified linear unit (ReLU) for activation to allow nonlinearity, which entails the neural network learning intricate patterns. A dropout layer with a rate of 0.5 follows next; this prevents overfitting. Overfitting may be averted through the random dropping of units during the training phase, along with their connections, forcing the neural network to attain a rugged approximation of the data. The last layer is dense_1, comprising a single neuron that applies the sigmoid activation function, and its output is a number ranging from to 0–1, representing the probability of the trained image belonging to one of the classes. This is how the model works; this output is based on the routines of the encoder and decoder to suggest the potential class of the image. The full process is a symphony of convolution, activation functions, pooling, pre-classification, and dense layers, one feeding up the knowledge to the next layer.

The architecture of this CNN consists of several layers, each with specific functions and parameters, as summarized in Table 3. This table provides a detailed overview of the CNN architecture, describing each layer’s type, output shape, number of trainable parameters, and functionality. The model begins with an input layer that processes images of size 150 × 150 pixels with three color channels (red, green, blue; RGB). It then sequentially applies convolutional and pooling layers to extract features and reduce spatial dimensions, culminating in fully connected and output layers designed for binary classification. Additionally, key components like dropout regularization are incorporated to mitigate overfitting, while the sigmoid activation function outputs probabilities for the target class.

Table 3 Architectural details of the CNN model.

Layer type and configuration	Output shape	Number of parameters	Description and functionality	
Input Layer	(None, 150, 150, 3)	0	Accepts input images with dimensions 150 × 150 pixels and three color channels (RGB).	
Convolutional Layer 1	(None, 148, 148, 32)	896	32 filters of size (3 × 3), Activation: ReLU. Extracts basic edge features.	
MaxPooling Layer 1	(None, 74, 74, 32)	0	Pool Size: (2 × 2). Reduces spatial dimensions by half.	
Convolutional Layer 2	(None, 72, 72, 64)	18,496	64 filters of size (3 × 3), Activation: ReLU. Captures higher-level features.	
MaxPooling Layer 2	(None, 36, 36, 64)	0	Pool Size: (2 × 2). Further downsampling for computational efficiency.	
Convolutional Layer 3	(None, 34, 34, 128)	73,856	128 filters of size (3 × 3), Activation: ReLU. Detects complex patterns.	
MaxPooling Layer 3	(None, 17, 17, 128)	0	Pool Size: (2 × 2). Compresses feature maps while retaining important information.	
Flatten Layer	(None, 36,992)	0	Converts 3D feature maps into a 1D vector for fully connected layers.	
Fully Connected Layer 1	(None, 512)	18,940,416	Dense Layer: 512 units, Activation: ReLU. Learns high-level abstract features.	
Dropout Layer	(None, 512)	0	Dropout Rate: 0.5. Mitigates overfitting by randomly deactivating nodes.	
Output Layer	(None, 1)	513	Dense Layer: 1 unit, Activation: Sigmoid. Produces binary classification output.	
Total Parameters:	19,034,177 (72.61 MB)	
Trainable Parameters:	19,034,177 (72.61 MB)	
Non-trainable Parameters:	0 (0.00 Byte)	

Table 4 complements this by detailing the training and compilation parameters utilized during model development. It specifies the data preprocessing techniques, such as rescaling and augmentation strategies, along with the optimizer, loss function, batch size, and performance metric. These parameters form the experimental setup, ensuring the reproducibility and robustness of the training process. Together, the two tables provide a comprehensive overview of the CNN’s architecture and the methodology employed for its training and evaluation.

Table 4 Training hyperparameters and compilation parameters used in the CNN model.

Parameter	Value	
Input image dimensions	150 × 150	
Batch size	64	
Image rescaling factor	1/255	
Data augmentation	Shear = 0.2, Zoom = 0.2	
Optimizer	Adam	
Loss function	Binary Crossentropy	
Performance metric	Accuracy	
Number of epochs	20	
Learning rate	0.0003	

Model evaluation

In this subsection, an inference engine incorporating deep learning classifications was implemented to predict stock price directional movements (upward or downward trends). The predictive performance of these models was evaluated using two established statistical metrics: Accuracy and F1 score. The accuracy metric is calculated as follows:

(2) Accuracy=TP+TNTP+TN+FP+FN.

TP represents the model prediction, which is also true, and the real sample is also true; TN and True Negatives when the model predicts false, the real sample is also false; FP and False Positives when the model predicts true, while the real sample is false; and FN is a false negative when the model predicts false, but the real sample is true. Precision is a measure of the accuracy of the positive samples in the prediction data and recall is a measurement of the coverage of the positive samples in the model’s prediction data. The formulas for Precision and Recall are as follows:

(3) Precision=TPTP+FP

(4) Recall=TPTP+FN.

F1 is used in statistical classification to evaluate the accuracy of a binary model derived from both precision and recall.

(5) F1=2×Precision×RecallPrecision+Recall.

In this study, our evaluation model considers not only the accuracy index but also the precision and recall indicators.

Experimental results

The experimental setup was configured using an Intel Core i7-6700HQ CPU with 4 cores and eight threads, paired with an NVIDIA GeForce GTX 950M GPU with 2 GB memory. The system included 16 GB of RAM and a 1TB HDD, providing a balanced environment for computational tasks and data storage. To facilitate reproducibility and allow for independent verification of the experimental results, the complete set of codes used in this study has been made publicly accessible on an open-access version control platform (https://github.com/Edreesrm/Enhancing-Market-Trend-Prediction-Using-Convolutional-Neural-Networks-).

In the methodology section, we delineate the procedures employed to generate sub-charts embodying JCs patterns to enhance analysis. The very first step was to identify the best window size in which we should train our CNN model because the window size is one of the most essential hyperparameters that has to be considered if one wants to capture meaningful patterns from the time-series data.

We considered a variety of window sizes; for instance, 5, 10, 15, 20, 25, and 30 for the best working size. The data that any window size models at any one time is different. It might be the case that the small window size is better at capturing fine-grained short-term trends and the large window size in catching general long-term ones. We also set a consistent amount of transformation size for all window configurations: the size was set to occupy half of the window size, thereby ensuring a 50% overlap between consecutive windows. This is important for maintaining coherence and continuity in the data, enabling the model to transit from one window to another without losing core information. This strengthens not only the ability of the model to generalize across different data segments but also helps avoid possible sharp changes that can occur, therefore disrupting the learning process.

We designed the dataset meticulously, placing significant focus on combining the carefully selected window sizes and strategic use of overlapping windows. This extensively compiled dataset represents the underlying temporal dynamics in JCs charts. These underlying very intricate patterns are essential in achieving accurate pattern recognition or prediction in our binary classification tasks. This comprehensive methodological approach paves the way for successful training and evaluation of our CNN models. Upon identification of a window containing candlestick patterns, the Technical Analysis Library (Ta-lib) was utilized to determine if any of the 61 predefined patterns were present. If a pattern was identified, then the window was selected for further analysis. The subsequent phase involves classifying the window as indicative of either a bullish or bearish trading pattern.

The classification of market trends is supported by various technical indicators, primarily focusing on the relationship between the price of the last candle in the window and moving average. Moving averages of 20, 50, and 200 periods were tested; however, for the purposes of this study, a 20-period moving average was selected. If the price of the last candle exceeded the moving average, the window was classified as exhibiting an uptrend trend. Conversely, if the price was below the moving average, this was indicative of a downtrend trend. This approach provides a robust framework for trend analysis by integrating candlestick pattern recognition with the key technical indicators. In the Forex market, the formation of trading patterns is a testament to its dynamic nature, in which each pattern emerges distinctly from its predecessor, introducing new configurations with each occurrence. This ever-evolving pattern landscape presents a considerable challenge for traders and analysts, making it difficult to track and analyze every pattern comprehensively. The realization that pattern mining within the forex market is an ongoing, never-finished process underscores the complexity and fluidity of market behavior.

The useful results of using TA-Lib in pattern analysis can be seen in Table 5 which show how often the patterns found with the library happened. The data provided in the table reflect the diversity and frequency of the encountered patterns. A table typically lists the names of the patterns alongside the number of times each pattern is identified within the dataset, providing a clear view of which patterns are the most common and potentially signaling-specific market conditions or trends. It has been noted that certain patterns become more prominent, whereas others become less visible.

Table 5 Frequency of various candlestick patterns within a 15-min timeframe.

Pattern	Occurrences	Pattern	Occurrences	
Spinning Top	19,678	Stalled Pattern	346	
Long Line Candle	19,058	Evening Star	340	
Belt-Hold	17,318	Three-Line Strike	337	
Short Line Candle	15,741	Identical Three Crows	250	
Closing Marubozu	14,445	Morning Doji Star	106	
Doji	14,133	Evening Doji Star	104	
Hikkake Pattern	11,894	Modified Hikkake Pattern	119	
High-Wave Candle	11,640	Thrusting Pattern	117	
Rickshaw Man	9,553	Three Advancing White Soldiers	86	
Engulfing Pattern	7,990	Piercing Pattern	76	
Marubozu	5,659	Dark Cloud Cover	68	
Harami Pattern	4,370	Homing Pigeon	47	
Three Outside Up/Down	3,854	On-Neck Pattern	32	
Hammer	2,776	Stick Sandwich	32	
Gravestone Doji	2,047	In-Neck Pattern	27	
Dragonfly Doji	1,942	Tristar Pattern	24	
Takuri (Dragonfly Doji with long lower shadow)	1,906	Three Black Crows	24	
Hanging Man	1,537	Tasuki Gap	19	
Matching Low	1,474	Unique 3 River	9	
Harami Cross Pattern	1,082	Ladder Bottom	7	
Doji Star	967	Breakaway	4	
Three Inside Up/Down	793	Abandoned Baby	1	
Upside/Downside Gap Three Methods	719	Two Crows	1	
Shooting Star	669	Counterattack	1	
Separating Lines	624			

In an analysis of the candlestick patterns at a 15-min interval in the historical forex dataset, certain patterns were not identified at all. These patterns are “Kicking,” “Kicking-bull/bear determined by the longer marubozu,” “Three Stars in the South,” “Concealing Baby Swallow,” “Mat Hold,” and “Upside Gap Two Crows.” The non-identification of these patterns deems that the specific characteristics of the market or of the period under study did not fulfill the necessary conditions for their occurrence. Candlestick patterns that rarely occurred in the dataset are presented in Table 5. They are rare because in normal market conditions, trends of these patterns appear very seldom but they could be very crucial in the context of possible major market trends or reversals.

In this study, we thoroughly explored how different window sizes and shifts impact the creation of image datasets for the CNN models used in binary classification. We tested six distinct configurations to see how various combinations of window sizes and shifts would affect a time-series dataset. First configuration: We started with a window size of 5 and a shift of 2, creating a dataset of 6,292 images divided into two classes. This dataset was split into 4,403 training images, 628 validation images, and 1,261 test images.

Second configuration: We used a window size of 10 and a shift of 5, generating 4,970 images, which were allocated as 3,478 for training, 496 for validation, and 996 for testing.

Third configuration: We applied a window size of 15 and a shift of 7, resulting in 11,130 images, distributed into 7,790 for training, 1,112 for validation, and 2,228 for testing.

Fourth configuration: Using a window size of 20 and a shift of 10, we produced 3,675 images, split into 2,572 for training, 366 for validation, and 737 for testing.

Fifth configuration: With a window size of 25 and a shift of 12, we created 6,541 images, divided into 4,578 training images, 653 validation images, and 1,310 test images.

Sixth configuration: Finally, a window size of 30 and a shift of 15 gave us 5,181 images, which were split into 3,626 for training, 517 for validation, and 1,038 for testing.

Each configuration was designed to see how different window sizes and shifts would affect the CNN’s ability to identify and classify patterns in the data. The data sets created through these methods were crucial for training, validating, and testing CNN models. They provided varied ways of representing temporal data, which influenced the model’s ability to generalize and perform on unseen data. This in-depth examination of window sizes and shifts lays the groundwork for optimizing data preparation strategies for CNNs, particularly for tasks that require detailed recognition of temporal patterns. The study also includes a detailed table detailing the resolution of each window. We constructed this table using the established methodological framework and systematically presented the window-specific results. Table 6 shows the results such a detailed breakdown facilitates a comprehensive understanding of how each window’s dimensions influence overall model performance, offering insights into the optimal configurations for future predictive modeling.

Table 6 The performance results according to common metrics obtained from the implemented model.

Window size	Shift size	Precision	Recall	F1-Score	Accuracy	
5	2	0.993	0.993	0.993	0.993	
10	5	0.977	0.977	0.977	0.977	
15	7	0.988	0.988	0.988	0.988	
20	10	0.969	0.969	0.969	0.969	
25	12	0.982	0.982	0.982	0.982	
30	15	0.974	0.974	0.974	0.974	

Figure 6 shows the performance of the CNN model during training. Each graph shows the trends of the accuracy and loss over successive epochs for both the training and validation datasets.

Figure 6 Loss function and accuracy of CNNs across different window sizes and shifts.

Figure 7 shows that the receiver operating characteristic (ROC) curve graphs indicate the performance of the classification model. In almost all cases, the ROC curves showed a near-perfect score with an area under the curve (AUC) of 1.00, signifying an almost classifier with a true positive rate across various thresholds and a negligible false positive rate.

Figure 7 ROC curve graphs for various window and shift sizes in the model’s evaluation.

For the performance of the CNN models, this study assessed how it varied given a change in the window sizes and shifts under a cross-validation binary classification task. We conducted experiments on six different configurations of window size and shift applied to a time-series dataset. Each configuration was tested using stratified k-fold cross-validation (k = 5) to avoid overestimation when reporting those performances. Table 7 presents a comprehensive comparative analysis of model performance across six distinct configurations, each implementing different window sizes (W) and shift parameters (S).

Table 7 Performance evaluation of loss and accuracy across multiple configurations and cross-validation folds.

Fold	Config 1: W = 5, S = 2	Config 2: W = 10, S = 5	Config 3: W = 15, S = 7	Config 4: W = 20, S = 10	Config 5: W = 25, S = 12	Config 6: W = 30, S = 15	
	Loss	Acc	Loss	Acc	Loss	Acc	Loss	Acc	Loss	Acc	Loss	Acc	
1	0.0293	0.9929	0.0284	0.9879	0.0170	0.9933	0.1002	0.9633	0.0481	0.9855	0.0385	0.9875	
2	0.0124	0.9944	0.0604	0.9829	0.1131	0.9816	0.0770	0.9714	0.0536	0.9801	0.0658	0.9817	
3	0.0374	0.9873	0.0461	0.9889	0.0428	0.9906	0.0240	0.9932	0.0753	0.9839	0.0845	0.9788	
4	0.0339	0.9913	0.0601	0.9789	0.0260	0.9924	0.0642	0.9796	0.0466	0.9870	0.0990	0.9624	
5	0.0140	0.9936	0.0659	0.9819	0.0256	0.9933	0.0503	0.9850	0.0643	0.9748	0.0636	0.9826	
Average	0.0254	0.9919	0.0522	0.9841	0.0449	0.9902	0.0631	0.9785	0.0576	0.9823	0.0703	0.9786	

The experimental results demonstrate varying degrees of effectiveness across configurations, with notable performance patterns emerging from the cross-validation analysis. Configuration 1 (W = 5, S = 2) exhibited exceptional performance metrics, achieving the highest mean accuracy of 0.9919 and the lowest average loss of 0.0254, with remarkably consistent performance across all five folds (accuracy range: 0.9873–0.9944). In a detailed analysis of Configuration 1, the model demonstrated outstanding stability across all cross-validation folds, with the second fold achieving the highest accuracy (0.9944) and the lowest loss (0.0124). In contrast, the third fold showed slightly lower accuracy (0.9873) with a marginally higher loss (0.0374). This exceptional performance stability across different data partitions indicates strong generalization capabilities and reliable model behavior under varying data conditions. Configuration 2 (W = 10, S = 5) showed a slight decrease in performance (mean accuracy: 0.9841, mean loss: 0.0522), while Configuration 3 (W = 15, S = 7) maintained robust performance with a mean accuracy of 0.9902 and mean loss of 0.0449. As window sizes increased, a gradual decline in performance was observed, with Configuration 4 (W = 20, S = 10) showing mean accuracy of 0.9785 and mean loss of 0.0631, Configuration 5 (W = 25, S = 12) achieving mean accuracy of 0.9823 and mean loss of 0.0576, and Configuration 6 (W = 30, S = 15) recording mean accuracy of 0.9786 and mean loss of 0.0703. This systematic evaluation reveals an inverse relationship between window size and model performance, suggesting that smaller window sizes with proportionally smaller shifts are more effective at capturing relevant temporal patterns while maintaining classification accuracy. The cross-validation results demonstrate the robustness of the model across all configurations, with accuracy consistently exceeding 0.96. However, larger window sizes generally corresponded to increased loss values and more significant variability between folds. The consistently low standard deviation in accuracy and loss metrics for Configuration 1 underscores its superior performance characteristics and establishes it as the optimal configuration for this specific application.

Table 8 presents a comprehensive summary of the model’s performance metrics across different configurations. The analysis encompasses six distinct configurations, each characterized by specific window sizes and shift parameters.

Table 8 The summary of average accuracy and loss for each configuration.

Configuration	Window size	Shift	Average loss	Average accuracy	
Config 1	5	2	0.0254	0.9919	
Config 2	10	5	0.0522	0.9841	
Config 3	15	7	0.0449	0.9902	
Config 4	20	10	0.0631	0.9785	
Config 5	25	12	0.0576	0.9823	
Config 6	30	15	0.0703	0.9786	

Configuration 1, with a window size of 5 and shift of 2, demonstrates the highest accuracy (0.9919) and lowest loss (0.0254) among all tested configurations. Configuration 3 (window size: 15, shift: 7) achieves the second-best performance with an accuracy of 0.9902 and loss of 0.0449. Notably, while larger window sizes (Configuration 4–6) maintain relatively high accuracy above 0.97, they exhibit incrementally higher loss values, with Configuration 6 (window size: 30, shift: 15) showing the highest loss at 0.0703.

The graphs in Fig. 8 show the training and validation performance metrics (accuracy and loss) at different times for the CNN model tested experimentally using the 5-fold k-fold cross-validation method. Each figure represents a different configuration or training configuration of the model.

Figure 8 Loss function and accuracy for different windows and shift sizes after using the cross-validation.

The receiver operating characteristic (ROC) curves were generated to evaluate the classification performance of various CNN model configurations, as illustrated in Fig. 9. The analysis, conducted using 5-fold cross-validation on time-series data, demonstrates robust discriminative capabilities across all configurations. Each configuration exhibits ROC curves approaching the optimal classification point (top-left corner of the plot), indicating high sensitivity and specificity ratios. The performance metrics reveal consistent model behavior across different window and shift parameters. The configurations demonstrate exceptional true positive rates while maintaining minimal false positive rates, suggesting robust binary classification performance. These results, visualized through ROC curves for six distinct configurations (window sizes ranging from 5 to 30 and corresponding shifts from 2 to 15), provide comprehensive insight into the model’s classification efficacy across varying temporal scales. Figure 9 illustrates the ROC curves obtained from cross-validation experiments, providing empirical evidence of the model’s consistent performance across different temporal window configurations.

Figure 9 ROC curve graphs for various window and shift sizes after using the cross-validation.

Table 9 presents a comprehensive comparative analysis of nine pre-trained architectures, evaluating their performance through precision, recall, F1-score, and accuracy metrics.

Table 9 Performance metrics of pre-trained CNN models.

CNN model	Precision	Recall	F1-score	Accuracy	
VGG19	0.927	0.926	0.925	0.925	
VGG16	0.925	0.925	0.925	0.925	
ResNet50	0.912	0.908	0.907	0.907	
MobileNet	0.935	0.933	0.933	0.933	
EfficientNetB0	0.922	0.921	0.920	0.920	
InceptionResNetV2	0.255	0.500	0.337	0.509	
MobileNetV2	0.907	0.907	0.907	0.907	
DenseNet121	0.863	0.809	0.804	0.812	
InceptionV3	0.690	0.592	0.528	0.586	

The empirical results demonstrate that MobileNet exhibits superior performance across all evaluation metrics, achieving the highest precision (0.935), recall (0.933), F1-score (0.933), and accuracy (0.933). The VGG architectures demonstrate robust performance, with VGG19 achieving precision of 0.927, recall of 0.926, and both F1-score and accuracy at 0.925, while VGG16 shows remarkable consistency with uniform metrics of 0.925 across all evaluation criteria. EfficientNetB0 maintains strong performance with metrics consistently above 0.920, followed by ResNet50 and MobileNetV2, which demonstrate comparable performance patterns with approximately 0.907 across all metrics. DenseNet121 exhibits moderate performance with an accuracy of 0.812 and recall of 0.809, while the Inception-based architectures demonstrate significantly lower performance compared to other models, with InceptionResNetV2 achieving only 0.509 accuracy and particularly low precision (0.255), and InceptionV3 showing suboptimal performance with accuracy at 0.586 and F1-score at 0.528. The performance analysis establishes a clear hierarchy among the evaluated models, with MobileNet emerging as the optimal choice, followed by the VGG family, EfficientNetB0, and the ResNet50/MobileNetV2 implementations, while the Inception-based architectures demonstrate limited effectiveness in the current application context.

Discussion

This section presents a comprehensive analysis of the proposed candlestick chart forecasting methodology, examining its distinctive characteristics, performance metrics, computational complexity, and real-world applications. The discussion is structured into three main subsections: Comparative Analysis, Model Complexity Evaluation, and Real-Time Market Implementation.

Comparative analysis with existing literature

A comparative analysis of the proposed candlestick chart forecasting methodology with existing literature reveals several key distinctions, as highlighted in Table 10. The methodology uniquely employs 61 Candlestick patterns with CNN, departing from conventional approaches that utilize various methodologies such as ensemble machine learning, deep learning models, graph neural networks, and optimization algorithms. For instance, Lin et al. (2021) implemented a combination of random forest, GBDT, LR, KNN, SVM, and LSTM, while Chen & Tsai (2022) utilized a YOLO model for dynamic pattern detection.

Table 10 The comprehensive comparison of the related studies using Candlestick charts.

Authors	Stock market	Time frame	Model used	Input dataset	Classification	Accuracy	
Lin et al. (2021)	China’s	Daily	RF, GBDT, LR, KNN, SVM, LSTM	Historical stock prices, technical indicators	Categorical	60%	
Hung & Chen (2021)	Taiwan and Tokyo	Daily	CNN-autoencoder, RNN	Candlestick Charts	Binary	84%	
Ho & Huang (2021)	Apple, Tesla, IBM, Amazon, and Google	Daily	1-D CNN and 2-D CNN	Candlestick charts + Twitter Text	Binary	75.38%	
Ardiyanti, Palupi & Indwiarti (2021)	IDX	Daily	ANN+K-Fold Cross-Validation	Candlestick Pattern data	Binary	85.96%	
Chen & Tsai (2022)	Foreign exchange (EUR/USD)	1-min	YOLO model	GAF encoded candlestick charts	Binary	88.35%	
Liang et al. (2022)	China’s	Daily	K-line	K-line patterns	Binary	56.04% and 55.56%	
Santur (2022)	11 world indices	Daily	Ensemble Learning-Xgboost	Candlestick Chart	Binary	53.8%	
Wang et al. (2022)	China’s (CSI 300)	Daily	Graph Neural Network	Candlestick is represented by graph embedding	Categorical	–	
Behar & Sharma (2022)	Indian (BSE and NIFTY 50) and US (S&P500 and DJIA)	Daily	KNN	Candlestick charts	Binary	61.4%	
Ramadhan, Palupi & Wahyudi (2022)	Nasdaq100	Hourly	CNN-LSTM	GAF encoded candlestick charts	Binary	90% and 93%	
Puteri et al. (2023)	Forex (GBP/USD)	4-h	SVM	OHLC candlestick data	Binary	90.72%	
Ruixun Zhang & Lin (2023)	Exchange-traded funds (ETF)	Daily	Channel and Spatial-Attention CNN (CS-ACNN)	Candlestick charts	Binary	Sharpe ratios between 1.57 and 3.03	
Vijayababu, Bennur & Vijayababu (2023)	Ahihi Dataset	Daily	VGG16, ResNet50, AlexNet, GoogleNet, YOLOv8	OHLC candlestick pattern	Binary	91.51%	
Chen, Hu & Xue (2024)	Chinese	Daily	Bidirectional GRU with Candlestick Patterns and Sparrow Search Algorithm (SSA-CPBiGRU)	OHLC candlestick data	Categorical	–	
Huang, Wang & Wang (2024)	Chinese (Kweichow Moutai, CSI 100, and 50 ETF)	Daily	Vector auto-regression (VAR), Vector error correction model (VECM)	OHLC candlestick data	Binary	–	
Proposed model	Forex (EUR/USD)	15-min	CNN	Candlestick charts	Binary	99.3%	

The present study’s focus on Forex market data with a 15-min interval represents a significant methodological distinction. This differs notably from existing research that employs diverse data sources, including China’s Stock Market Data (CSI 300 and CSI 500 indices), Taiwan Stock Exchange, Nikkei 225, social media sentiment data (Twitter-based analysis for Apple, Tesla, IBM), hourly Nasdaq100 time series data, and ETFs from various markets. The implemented 15-min interval framework provides high-frequency insights, contrasting with the daily or hourly intervals employed in other studies, such as those by Lin et al. (2021), Behar & Sharma (2022), and Wang et al. (2022), the 1-min intervals in the Dynamic Deep Convolutional Candlestick Learner study, and the 4-h intervals in the support vector machine study for Foreign Exchange prediction. The proposed methodology exclusively utilizes CNNs, whereas existing studies employ a broader range of models including ensemble learning models (RF, GBDT, Xgboost), deep learning architectures (CNN-autoencoder, RNN, CS-ACNN, VGG16, ResNet50, AlexNet, GoogleNet, YOLOv8), and optimization algorithms (Sparrow Search Algorithm with BiGRU). Additionally, while the present study focuses on OHLC candlestick data, other research incorporates diverse datasets including historical stock prices, technical indicators, sentiment analysis data combined with candlestick charts, and graph embeddings representing candlestick data. The methodology achieves a superior accuracy rate of 99.3%, significantly outperforming existing models that typically range between 56% and 91.51%. Specifically, the Dynamic Deep Convolutional Candlestick Learner achieved 88.35%, the support vector machine for Predicting Candlestick Chart Movement reached 90.72%, the Interpretable Image-Based Deep Learning study reported Sharpe ratios between 1.57 and 3.03, and the Enhancing Financial Chart Analysis study achieved 91.51%. These distinctions highlight the methodology’s superior accuracy, focused timeframe, and model simplicity through exclusive CNN utilization. This approach contrasts with more complex hybrid models employed in existing studies, such as combinations of autoencoders, RNNs, and advanced ensemble learning techniques. Furthermore, the specialized focus on the Forex market, distinct from analyses of stock indices, ETFs, or broader financial assets, combined with the comprehensive analysis of 61 distinct candlestick patterns, contributes to the enhanced accuracy.

Model complexity evaluation

A comprehensive complexity analysis of the proposed CNN architecture is presented in Table 11. The architectural analysis demonstrates an optimized structure comprising 19,034,177 parameters and requiring 221,008,385 FLOPs (Floating Point Operations), while maintaining an efficient memory utilization of 72.65 MB. The implemented five-layer architecture establishes an effective equilibrium between model depth and computational efficiency. The architectural configuration incorporates three convolutional (Conv2D) layers, systematically followed by maximum pooling (MaxPooling2D) layers, a flattened layer facilitating dimensional reduction, and two dense layers interconnected by an intermediate dropout layer for regularization purposes. A quantitative assessment of layer-wise computational characteristics indicates that the second convolutional layer exhibits the highest computational demands, requiring 95,883,264 FLOPs. In contrast, the primary dense layer encompasses the majority of model parameters, precisely 18,940,416. The implemented MaxPooling2D layers demonstrate significant efficacy in dimensional reduction while maintaining critical feature preservation, thereby enhancing the model’s computational efficiency. Performance evaluation through inference time analysis validates the architecture’s applicability for real-time implementations, exhibiting a mean processing latency of 570.20 ms. The observed temporal performance maintains consistency within an operational range of 397.73 ms to 741.64 ms, with a standard deviation of 84.14 ms.

Table 11 CNN model complexity analysis.

Model overview	
Total Parameters:	19,034,177	
Total FLOPs:	221,008,385	
Parameter Memory:	72.61 MB	
Model Size:	72.65 MB	
Model Depth:	5 Layers	
Layer-wise analysis	
Layer type	Parameters	FLOPs	Output shape	
Conv2D	896	19,625,984	(64, 148, 148, 32)	
MaxPooling2D	0	700,928	(64, 72, 72, 32)	
Conv2D	18,496	95,883,264	(64, 72, 72, 64)	
MaxPooling2D	0	331,776	(64, 36, 36, 64)	
Conv2D	73,856	85,377,536	(64, 34, 34, 128)	
MaxPooling2D	0	147,968	(64, 17, 17, 128)	
Flatten	0	0	(64, 36,992)	
Dense	18,940,416	18,940,416	(64, 512)	
Dropout	0	0	(64, 512)	
Dense	513	513	(64, 1)	
Inference time analysis	
Mean Inference Time:	570.20 ms	
Std Inference Time:	84.14 ms	
Min Inference Time:	397.73 ms	
Max Inference Time:	741.64 ms	

Real-time market analysis and results

Figure 10 presents the results of an algorithm tested on real-life market data for EUR/USD parity trend analysis. The price movements and directional signals between October 28, 12:00, and November 1, 00:00, 2024, were obtained under live market conditions. The graph indicates Trend reversal points by ‘U’ (Up) and ‘D’ (Down) markers. During the 84 h, the EUR/USD parity exhibited fluctuations within the approximately 1.078–1.088 range. A significant downward movement was observed around October 29, 12:00, where the price declined to approximately 1.078, followed by a recovery phase. The parity subsequently demonstrated an upward trend, reaching around 1.086–1.088. Analysis results demonstrate that the proposed trend detection algorithm generates effective signals under actual market conditions, particularly during periods of high volatility. The findings suggest potential improvements in financial market risk management strategies. Early detection of trend reversal points has been identified as crucial for institutional investors and portfolio managers’ position management decisions. The accuracy rate of signals detected in real-time data demonstrates higher sensitivity than traditional technical analysis methods. These findings highlight the future potential of artificial intelligence-enhanced trading systems. Further research opportunities include testing the algorithm across various periods and financial instruments to enhance the model’s generalizability.

Figure 10 Algorithmic trend detection signals on EUR/USD parity over 84-h period with 15-min intervals (October 28–November 1, 2024).

Conclusion

This research advances the prediction of financial market movements by integrating traditional JCs patterns from 17th-century Japanese trading practices with contemporary CNN. The study introduces a comprehensive methodological framework comprising three distinct stages, thereby establishing a novel approach to financial market analysis, particularly in the EUR/USD forex market context. The first stage implements an innovative sliding window technique for systematic time series analysis, facilitating the creation of detailed sub-graphs and the comprehensive analysis of 61 distinct candlestick patterns. This significantly expands upon traditional approaches, providing a more nuanced understanding of market dynamics. The second stage utilizes the Ta-lib library for automated pattern detection within sub-graph windows, complemented by multiple technical indicators for trend validation and an adaptive window-shift mechanism with six temporal configurations. The third stage implements a sophisticated CNN architecture, comprising 19,034,177 parameters and requiring 221,008,385 FLOPs, demonstrating remarkable computational efficiency with a mean inference time of 570.20 ms. The experimental results validate the effectiveness of this integrated approach, achieving an exceptional 99.3% prediction accuracy, substantially outperforming existing methodologies that typically range between 56% and 91.51%. This superior performance is further substantiated through rigorous 5-fold cross-validation on 15-min interval data and successful real-time market testing conducted between October 28 and November 1, 2024. The model’s robust performance in examining new and unseen data demonstrates its practical applicability, particularly in high-frequency trading environments where traditional pattern recognition methods often face limitations due to market volatility and noise.

The study’s methodological contributions advance theoretical understanding and practical implementation through several key innovations: a comprehensive pattern recognition system incorporating 61 candlestick patterns, an adaptive window-shift mechanism optimized for high-frequency data analysis, and a quantitative trend identification framework integrating SMAs. These advancements provide significant implications for automated trading systems and risk management strategies.

Future research directions can be categorized into several key areas. First, the model’s adaptability should be investigated across different financial instruments (e.g., cryptocurrencies, commodities) and various market conditions (high volatility, crisis periods), potentially leading to a more generalized prediction framework. Second, integrating additional technical indicators and fundamental analysis methods could enhance the model’s predictive capabilities, particularly by incorporating sentiment analysis and macroeconomic indicators. Third, the CNN architecture could be optimized further through neural architecture search and pruning methods, potentially reducing computational complexity while maintaining or improving accuracy. Fourth, the investigation of unsupervised and semi-supervised learning methodologies presents an opportunity to expand pattern recognition beyond predefined technical indicators. This approach may reveal previously unidentified market patterns and enhance the model’s analytical capabilities. Furthermore, the development of comprehensive performance evaluation frameworks incorporating multiple risk metrics and transaction cost analysis across various market conditions would strengthen the model’s practical validation. Fifth, the temporal aspect of the analysis could be expanded to include longer time horizons and multiple time-frame analysis, providing a more comprehensive market view. Finally, developing adaptive learning mechanisms that can automatically adjust to evolving market conditions represents a promising direction for enhancing the model’s long-term effectiveness.

In conclusion, this research significantly advances the field of financial market prediction by successfully synthesizing traditional technical analysis methods with modern artificial intelligence techniques. The demonstrated capability to accurately predict market trends using this comprehensive approach contributes to the growing body of literature on artificial intelligence applications in financial markets and provides practical tools for market participants in their decision-making processes. Integrating historical trading wisdom with cutting-edge deep learning technology establishes a robust framework for future developments in automated trading systems and market analysis tools, marking a significant step forward in the evolution of financial forecasting methodologies.

Limitations and validity

The dataset utilized in this research was sourced from the Forex website and encompassed various trading intervals. Quarter-hourly data containing approximately 100,000 price points was selected. This data is converted into JC charts to predict price changes during trading sessions. While this study presents a novel approach to financial market prediction by integrating JC patterns and CNN architecture, several limitations should be acknowledged to correctly interpret the results and future research considerations.

Data-related limitations

Significant variability across experimental configurations complicates performance evaluation.

Imbalanced sample distribution among training, validation, and test sets potentially affects model generalization.

Limited representation of diverse market conditions, excluding economic indicators, market sentiment, and news events.

Despite comprehensive quarter-hourly data coverage (approximately 100,000 price points), the dataset may not fully capture all market dynamics and behavioral patterns.

Conversion process from raw data to JCs charts might introduce information loss or transformation artifacts.

Methodological limitations

Window and shift sizes determined empirically rather than through optimization.

SMA usage for trend classification introduces delayed responses and potential lag in signal generation.

Reliance on predefined patterns in the Talib library restricts identification of emerging or novel pattern combinations.

Limited integration of advanced technical indicators may omit valuable market insights.

Technical and implementation challenges

High computational demands for large dataset processing and complex model training.

Market-specific training may limit applicability to different financial instruments and market conditions.

Potential scalability issues for real-time applications and multiple market analysis.

Integration challenges with existing trading systems and latency concerns in high-frequency trading environments.

To address these limitations, the study implemented several validity measures including cross-validation techniques, multiple evaluation metrics, real-market testing, and comparative analysis with existing methodologies. These measures provide context for result interpretation and guide future research directions while ensuring the methodology’s practical utility in various market conditions.

Supplemental Information

Supplemental Information 1 Bullish Candlestick Patterns: Structural examples and components of candle types.

Supplemental Information 2 Bearish Candlestick Patterns: Structural examples and components of candle types.

Supplemental Information 3 Architectural Details of the CNN Model.

Supplemental Information 4 Training Hyperparameters and Compilation Parameters Used in the CNN Model.

Supplemental Information 5 Frequency of various candlestick patterns within a 15-minute timeframe.

Supplemental Information 6 The performance results according to common metrics obtained from the implemented model.

Supplemental Information 7 Performance Evaluation of Loss and Accuracy Across Multiple Configurations and Cross-Validation Folds.

Supplemental Information 8 The summary of Average Accuracy and Loss for Each Configuration.

Supplemental Information 9 Performance Metrics of Pre-trained CNN Models.

Supplemental Information 10 The comprehensive comparison of the related studies using Candlestick Charts.

Supplemental Information 11 CNN Model Complexity Analysis.

Additional Information and Declarations

Competing Interests

The authors declare that they have no competing interests.

Author Contributions

Edrees Ramadan Mersal conceived and designed the experiments, performed the experiments, performed the computation work, prepared figures and/or tables, and approved the final draft.

Kürşat Mustafa Karaoğlan conceived and designed the experiments, analyzed the data, prepared figures and/or tables, authored or reviewed drafts of the article, and approved the final draft.

Hakan Kutucu conceived and designed the experiments, analyzed the data, performed the computation work, authored or reviewed drafts of the article, and approved the final draft.

Data Availability

The following information was supplied regarding data availability:

The third-party raw data is available at Forex Software Ltd: https://forexsb.com/historical-forex-data.

The code is available at GitHub and Zenodo:

- https://github.com/Edreesrm/Enhancing-Market-Trend-Prediction-Using-Convolutional-Neural-Networks-

- Ramadan, E., & Mersal. (2024). Enhancing-Market-Trend-Prediction-Using-Convolutional-Neural-Networks. Zenodo. https://doi.org/10.5281/zenodo.14201499.

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
