# Peer review of "Enhancing market trend prediction using convolutional neural networks on Japanese candlestick patterns"

_PeerJ Computer Science, doi:10.7717/peerj-cs.2719_

## Round 0.1 · original submission · Major Revisions

Applying deep learning techniques to financial chart patterns is an approach that offers potential for both market analysis and investment decisions. However, addressing the peer reviewer concerns will be important before publication.

Reviewer 1 ·

Basic reporting

I started reviewing your paper as one that structurally excited me. However, upon review, I could only find the numerical and verbal results of the achievements described and realized. In your paper, no actual test application on a real graph has been provided. For example, in YOLO object detection, has the dog been correctly predicted? What is the prediction percentage? This is typically provided in every updated paper, but in your case, I looked for the response of your developed method on a candlestick chart, but I couldn't find it. We would like to see the actual formation of percentage achievements on real data. It would be more appropriate to make an evaluation based on that.

Experimental design

Appropriate

Validity of the findings

To discuss the validity of the findings, it is necessary to see the model's actual detections.

·

Basic reporting

Overall, this paper presents an interesting approach by leveraging Convolutional Neural Networks (CNNs) to analyze Japanese candlestick (JC) charts for predicting future price movements in financial markets. The attempt to apply deep learning techniques to financial chart patterns is promising and innovative. However, there are several key issues and clarifications required for the paper to be more transparent and scientifically sound.

Experimental design

1. Number of Pixels in Charts
Given the large input size of 150x150 pixels, the flattened vector would indeed be very large, which can increase the complexity of the model and the number of parameters in the fully connected layers. If I am not mistaken, after the final convolution and pooling layers, the feature map is reduced to 17x17 (as mentioned in the architecture). In the last convolutional layer, there are 128 filters. The size of the feature map after the final pooling layer would be: 17×17×128=36,992 and 17×17×128=36,992. This means the flattened vector would have 36,992 elements before being fed into the fully connected layer. Feeding such a large vector into the fully connected layer with 512 neurons means there would be a significant number of learnable parameters in the fully connected layers. The number of parameters between the flattened layer and the first fully connected layer would be: 36,992×512=18,926,464 parameters
And 36,992×512=18,926,464 parameters. This number excludes the bias terms, which would add an additional 512 parameters. With such a large number of parameters, there is a risk of overfitting, especially if the dataset is not large enough to support such complexity.

2. Computational Complexity
With the fully connected layer alone having close to 19 million parameters (just from the flattened vector to the first fully connected layer), the CNN model will need substantial training data to properly learn and generalize from these parameters. Additionally, the convolutional filters in the earlier layers also add to the number of parameters to be learned. For example, with 32, 64, and 128 filters in the convolutional layers, each filter is learning weights for the 3x3 patches it processes. This adds complexity to the model, increasing the data requirement. With such a large input size and many layers, the model might become computationally expensive and slow to train. Training deep models on high-dimensional data can be very resource-intensive, especially if the dataset is large. The authors should clarify if they encountered any issues with training time or memory usage, and if they applied techniques to reduce computational load (e.g., reducing input size or batch normalization).

3. Sufficiency of Charts for CNN Analysis
Without enough training data, the model may memorize specific features from the training charts, leading to overfitting. This means the model would perform well on the training set but poorly on unseen data (out-of-sample testing). To the best of my knowledge from the paper, I can not find evidence that authors used enough chart images to support this heavy learning process.

4. Clarity on RSI Signal Usage
The paper briefly mentions the use of the Relative Strength Index (RSI) as a technical indicator alongside candlestick patterns and SMA (Simple Moving Average). However, the explanation of how RSI signals are used for labeling or in model prediction is unclear. For example, it is not specified whether the RSI was used directly in the CNN model or simply as a confirmatory indicator. Further clarification on how RSI is integrated into the prediction process is needed.

Validity of the findings

1. Accuracy of 99.3% and Overfitting Concern
The reported accuracy of 99.3% seems unusually high for financial time-series data, where such performance is difficult to achieve due to inherent market noise and volatility. There is a strong possibility of “overfitting”, especially if the reported accuracy is in-sample. I recommend a more thorough analysis of overfitting risks, such as by using regularization techniques or model complexity controls.

2. Out-of-Sample Testing
The paper does not clearly state whether out-of-sample testing was conducted. If out-of-sample testing was performed, the authors should specify the exact time period used for testing. This is crucial for evaluating the model's ability to generalize to unseen market conditions.

Additional comments

This paper offers a compelling approach to candlestick chart analysis using CNNs, but several critical details are missing, or possibly the size of the model is too large, making it difficult to fully assess the methodology and results. I encourage the authors to address the aforementioned points to improve the clarity and robustness of their study.

Reviewer 3 ·

Basic reporting

.

Experimental design

.

Validity of the findings

.

Additional comments

Dear Authors,

Your paper is well motivated and clearly written. However, to further improve it and make it suitable for publication, please address the following suggestions:

1- Emphasize Contribution and Simulation Verification: Clearly emphasize the contribution of your work in relation to existing solutions in the literature, including supported simulation verification. This will highlight the novelty and significance of your research.

2- State Limitations: Clearly state the main difficulties and limitations of the proposed method in practical applications. Understanding the constraints will provide a balanced view of the method's applicability.

3- Update Literature Review: Most references are from before 2019. For such a popular and attractive topic, recent papers should be investigated.

4- Clarify Use of convolutional neural networks and Japanese candlestick patterns: The use of the Japanese candlestick patterns and convolutional neural networks is vague. Please explore and explain it in more detail to ensure readers understand its role and significance in your methodology.

5- Derive Complexity: The complexity of the proposed work should be derived. Providing a complexity analysis will help in understanding the computational feasibility of the method.

6- Revise Conclusion: The conclusion should summarize the research results and provide suggestions for future development prospects. Explain clearly and in detail, and revise this section to include these elements.

7- Summarize Literature: Include a table that summarizes the literature and highlights the contribution of your paper. This will help readers quickly grasp the context and significance of your work.

8- Expand Introduction: The introduction section seems weak. It must be expanded to include more details on the motivation and novelty of the paper. Clearly articulate what makes your work unique and why it is important.

9- Include Managerial Insights: Managerial insights are missing. Identify who the decision-makers in the study are and who will gain the maximum benefit from the proposed method. Providing this context will enhance the practical relevance of your research.
By addressing these points, your paper will be strengthened and more aligned with the journal's standards for publication.

Reviewer 4 ·

Basic reporting

The methods in the paper are generally clear and fairly robust, with a well-defined process for data preparation, pattern recognition, and model training.

But sometimes the paper shifts between present and past tenses, which can be improved to ensure consistency.

Overall, the paper is well-referenced, and the sources are correctly listed and cited.
In some instances, especially within the methodology section, it would help to further strengthen citations related to specific techniques, such as sliding window methodologies and hyperparameter tuning.

While related studies are referenced, outlining what previous research has lacked or where this study provides a novel contribution would strengthen the motivation for the research.

Further emphasis on literature gaps and potential future directions could enhance the clarity of its contributions and broaden its impact.

Experimental design

The methodology section of the paper provides a reasonably detailed explanation of the processes used.

However, there is a lack of specific details regarding the data cleaning process, how missing values were handled, or any specific filtering criteria applied.

The paper does not provide information on how hyperparameters (e.g., learning rate, batch size, number of epochs) were selected or tuned.

While the use of cross-validation is mentioned, the paper could provide more detail on how the folds were constructed (e.g., random splitting vs. stratified sampling).

The authors mention that CNNs work well for recognizing visual patterns, but they do not delve deeply into why CNNs are better suited for time-series data compared to other architectures like LSTM or SVM, which are traditionally used for such tasks.

Validity of the findings

The authors compare their CNN model with other machine learning models, indicating an effort to assess the relative performance of their approach.

But, the paper could include more context on why specific models were chosen for comparison and how the performance of these models was evaluated.

Additional comments

Overall, the paper maintains a clear and professional tone suitable, it is well-written and significant.

The combination of a comprehensive set of candlestick patterns, deep learning, and technical indicators represents a valuable advancement in market trend prediction.

---

## Round 0.2 · Major Revisions

Reviewer 2 has critical concerns about the article. These concerns need to be addressed.

·

Basic reporting

The authors’ CNN procedure and architecture are methodologically sound and adhere to established practices in machine learning for image classification. The reported accuracy, reaching up to 99.3%, effectively demonstrates the model’s ability to replicate predefined candlestick patterns identified by the TA-Lib library.

Initially, I assumed the authors’ aim was to extract predictive information from candlestick chart images, consistent with the overarching goal of candlestick analysis in financial forecasting. However, the study's objective is to develop a system for automatically identifying technical patterns already recognized within the TA-Lib library. The authors utilized these predefined patterns to label candlestick chart images. Patterns meeting predefined criteria, such as rising SMAs or specific candlestick formations, were labeled as either bullish or bearish. A CNN was subsequently trained to classify these images into the corresponding categories (bullish or bearish) based on the labeled dataset. The result is a machine learning system that accurately identifies and classifies predefined patterns in unseen candlestick chart images. The authors’ work effectively automates the process of recognizing established candlestick patterns, providing a consistent and scalable classification framework without manual intervention. This system achieves its intended purpose but remains limited to replicating existing technical analysis rules rather than uncovering new predictive insights or evaluating the financial utility of these patterns.

Experimental design

Comment 1: Reliance on Predefined Labels: The experimental design relies entirely on predefined technical patterns from the TA-Lib library for labeling. While this ensures consistency, it restricts the model’s ability to discover new or previously unrecognized patterns.
Suggestion: Incorporate unsupervised or semi-supervised learning techniques to explore whether the model can identify novel patterns that may hold predictive value.

Comment 2: Limited Real-World Testing: While the authors test the model on real-time EUR/USD data, they do not evaluate whether the CNN’s classifications translate into actionable financial outcomes (e.g., profitability or trading performance).
Suggestion: Include a performance analysis that connects the predicted bullish/bearish signals to actual market returns or trading strategies to validate practical utility.

Validity of the findings

The findings presented in the study demonstrate the effectiveness of the proposed CNN-based approach for accurately classifying candlestick chart patterns into predefined categories (bullish or bearish). However, while the reported 99.3% accuracy highlights the model's technical success in pattern recognition, the validity of these findings in terms of practical and financial relevance requires careful consideration.

Strengths Supporting Validity:
A: High Classification Accuracy: The reported accuracy of 99.3% reflects the model’s strong ability to replicate the predefined candlestick patterns identified using the TA-Lib library. The results are supported by rigorous cross-validation, ensuring that the high accuracy is not simply a result of overfitting to the training dataset.
B: Robust Testing Framework: The use of k-fold cross-validation and multiple experimental configurations (e.g., varying window sizes and shifts) adds credibility to the findings, as the results are derived from repeated evaluations on different subsets of the data.

Limitations Affecting Validity:
The study's reliance on TA-Lib's predefined candlestick patterns confines the model to classifying patterns based on established technical analysis rules derived from human recognition. While CNNs are powerful tools for detecting hidden and complex patterns that may not be immediately apparent to human observers, labeling predefined technical patterns does not fully leverage this capability. Instead, it serves as a systematic replication of existing human-defined frameworks. The true advantage of CNNs lies in their ability to uncover hidden predictive patterns in financial data, which could offer novel insights and an edge in market forecasting beyond traditional methodologies.

Reviewer 3 ·

Basic reporting

Dear Authors,

Your paper is well motivated and clearly written. However, to further improve it and make it suitable for publication, please address the following suggestions:

1- Emphasize Contribution and Simulation Verification: Clearly emphasize the contribution of your work in relation to existing solutions in the literature, including supported simulation verification. This will highlight the novelty and significance of your research.

2- State Limitations: Clearly state the main difficulties and limitations of the proposed method in practical applications. Understanding the constraints will provide a balanced view of the method's applicability.

3- Update Literature Review: Most references are from before 2019. For such a popular and attractive topic, recent papers should be investigated.

4- Clarify Use of neural networks and market trend prediction: The use of the market trend prediction and neural networks is vague. Please explore and explain it in more detail to ensure readers understand its role and significance in your methodology.

5- Derive Complexity: The complexity of the proposed work should be derived. Providing a complexity analysis will help in understanding the computational feasibility of the method.

6- Revise Conclusion: The conclusion should summarize the research results and provide suggestions for future development prospects. Explain clearly and in detail, and revise this section to include these elements.

7- Summarize Literature: Include a table that summarizes the literature and highlights the contribution of your paper. This will help readers quickly grasp the context and significance of your work.

8- Expand Introduction: The introduction section seems weak. It must be expanded to include more details on the motivation and novelty of the paper. Clearly articulate what makes your work unique and why it is important.

9- Include Managerial Insights: Managerial insights are missing. Identify who the decision-makers in the study are and who will gain the maximum benefit from the proposed method. Providing this context will enhance the practical relevance of your research.

By addressing these points, your paper will be strengthened and more aligned with the journal's standards for publication.

Experimental design

no comment

Validity of the findings

no comment

Additional comments

no comment

Reviewer 4 ·

Basic reporting

No comment

Experimental design

No comment

Validity of the findings

no comment

Additional comments

The article provides an innovative methodology presenting a modern approach to financial trend prediction.
The authors have performed a comprehensive evaluation and the article is well-structured.
The article also provides well-validated results ensuring robustness and reliability of the proposed model's performance.

---

## Round 0.3 · Major Revisions

Dear Authors,
Your submission stands out as an interesting study and has the potential to make a significant contribution to your field. However, to further exploit the strengths of CNNs as noted in the peer review, you might consider allowing your model to produce more complex outputs and discussing an approach that predicts future price direction (up/down) based on learned patterns in the data. Such a study could further highlight the capabilities of CNNs while increasing the scientific value of your work.
Yours sincerely.

·

Basic reporting

While I commend the authors' effort to integrate deep learning, particularly Convolutional Neural Networks (CNNs), with candlestick charts, I believe the current implementation does not fully utilize the strengths and capabilities of CNNs.

Experimental design

CNNs are highly effective at identifying subtle, non-linear relationships and patterns beyond explicitly defined rules or visual signals recognizable by humans. This unique capability makes CNNs particularly valuable for discovering patterns that are not immediately apparent or interpretable through traditional methods. However, in your current setup, the CNN is being used to classify technical signals (bullish or bearish) that are already well-defined and straightforward for humans to recognize.

Validity of the findings

As long as the CNN’s task is limited to predicting bullish or bearish classifications derived from technical signals, it is unsurprising to achieve high accuracy. This is because such tasks are relatively easy for humans as well, given the systematic and explicit nature of the rules used to generate the labels.

Additional comments

To truly leverage the power of CNNs, I recommend exploring settings where the model predicts more complex outcomes, such as the future price direction (up or down) based on learned patterns in the data. This would align more closely with the strengths of CNNs and provide greater value by addressing challenges that are not easily solvable by humans or rule-based systems.

---

## Round 0.4 · accepted · Accept

After a thorough and constructive review process, I am pleased to recommend the article for publication.

·

Basic reporting

Clarity and Organization
• The manuscript is now generally well-written and structured. The introduction and methodology are coherent and easy to follow.
• The updated figures and tables illustrate the main results effectively. Consider refining figure captions to ensure self-contained clarity.

References and Literature
• The authors have reasonably contextualized their study within existing literature (e.g., LSTM, CNN-based financial market prediction).
• Some newly added references are helpful to show the state of the art in candlestick pattern recognition.

Language and Formatting
• The overall language flow has improved since the initial draft. Minor proofreading for grammatical consistency is still suggested but not critical.
• Section labeling and headings are appropriately used.

Experimental design

Methodology and Procedures
• The authors describe how they collect data (EUR/USD 15-minute intervals), apply the sliding window, and label each window with SMA-based rules and candlestick patterns.
• Including the details of the CNN architecture (layers, hyperparameters) is appreciated.

Novel Aspects
• The approach of combining candlestick patterns (61 patterns from the TA-Lib library) with CNN-based classification is interesting and more comprehensive than typical small-pattern analyses.
• The multi-scale window approach (5-30 steps) is robust, helping capture different time horizons in Forex data.

Validity of the findings

Strengths
• The reported 99.3% accuracy is exceptionally high, supported by stratified 5-fold cross-validation.
• The out-of-sample simulation showing a net profit of 22.51% (after expenses) suggests real trading utility.

Remaining Concerns
• As previously noted, because the classification labels are partly derived from an SMA-based rule, there remains a question as to whether the CNN is primarily detecting the same SMA signals or truly uncovering deeper market structure.
• A direct baseline comparison (e.g., comparing the CNN’s performance to a simple moving-average cross strategy over the same out-of-sample period) would bolster the claim that the model captures more subtle patterns.
• Nonetheless, the extra layer of candlestick pattern detection plus CNN feature extraction suggests the model has more capability than simple heuristic approaches.

Overall Assessment of Validity
• The authors’ methodology is systematic, and they have provided additional clarifications in this round regarding their approach, cross-validation, and out-of-sample testing.
• While a formal baseline comparison would further strengthen the validity, the overall approach is sufficiently explained to conclude that the findings have merit.

Additional comments

The responses in this third round have meaningfully addressed most early concerns, including clarifying how the CNN processes multiple indicators and candlestick shapes. Although I still encourage a future direct comparison to simpler technical trading signals (e.g., MA crosses alone), the paper in its current form is acceptable.
The authors may wish to offer a short dedicated discussion or appendix in a future version/paper, contrasting “CNN vs. standard moving average trading results.” Doing so would help quantify the additive value of the CNN approach and preempt readers’ skepticism about potential overfitting to SMA-based labels.